# Incremental Scene Synthesis

**Benjamin Planche**[1,2]    **Xuejian Rong**[3,4]    **Ziyan Wu**[4]    **Srikrishna Karanam**[4]
**Harald Kosch**[2]    **YingLi Tian**[3]    **Jan Ernst**[4]    **Andreas Hutter**[1]

[1]Siemens Corporate Technology, Munich, Germany
[2]University of Passau, Passau, Germany
[3]The City College, City University of New York, New York NY
[4]Siemens Corporate Technology, Princeton NJ
{first.last}@siemens.com, {xrong,ytian}@ccny.cuny.edu, harald.kosch@uni-passau.de

## Abstract

We present a method to incrementally generate complete 2D or 3D scenes with the following properties: (a) it is globally consistent at each step according to a learned scene prior, (b) real observations of a scene can be incorporated while observing global consistency, (c) unobserved regions can be hallucinated locally in consistence with previous observations, hallucinations and global priors, and (d) hallucinations are statistical in nature, *i.e.*, different scenes can be generated from the same observations. To achieve this, we model the virtual scene, where an active agent at each step can either perceive an observed part of the scene or generate a local hallucination. The latter can be interpreted as the agent's expectation at this step through the scene and can be applied to autonomous navigation. In the limit of observing real data at each point, our method converges to solving the SLAM problem. It can otherwise sample entirely imagined scenes from prior distributions. Besides autonomous agents, applications include problems where large data is required for building robust real-world applications, but few samples are available. We demonstrate efficacy on various 2D as well as 3D data.

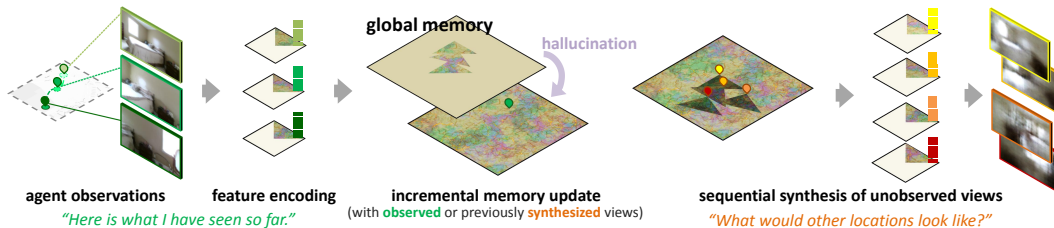

Figure 1: **Our solution** for scene understanding and novel view synthesis, given non-localized agents.

## 1 Introduction

We live in a three-dimensional world, and a proper cognitive understanding of its structure is crucial for planning and action. The ability to anticipate under uncertainty is necessary for autonomous agents to perform various downstream tasks such as exploration and target navigation [3]. Deep learning has shown promise in addressing these questions [31, 16]. Given a set of views and corresponding camera poses, existing methods have demonstrated the capability of learning an object's 3D shape via direct 3D or 2D supervision.

*Novel view synthesis* methods of this type have three common limitations. First, most recent approaches solely focus on single objects and surrounding viewpoints, and are trained with category-

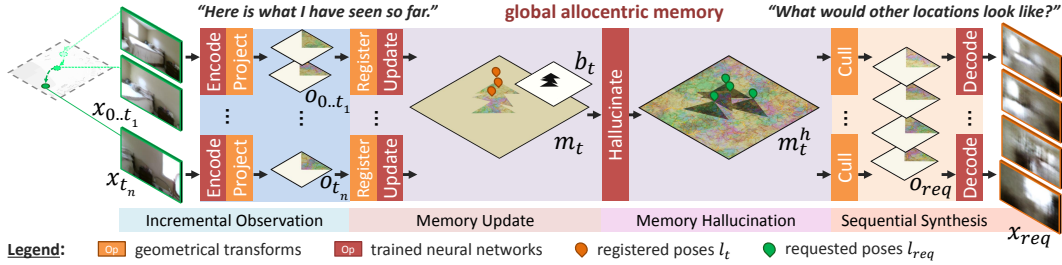

Figure 2: **Proposed pipeline** for non-localized agents exploring new scenes. Observations $x_t$ are sequentially encoded and registered in a global feature map $m_t$ with spatial properties, used to extrapolate unobserved content and generate consistent novel views $x_{req}$ from requested viewpoints.

dependent 3D shape representations (*e.g.*, voxel, mesh, point cloud model) and 3D/2D supervision (*e.g.*, reprojection loss), which are not trivial to obtain for natural scenes. While recent works on auto-regressive pixel generation [22], appearance flow prediction [31], or a combination of both [21] generate encouraging preliminary results for scenes, they only evaluate on data with mostly forwarding translation (*e.g.*, KITTI dataset [9]), and no scene understanding capabilities are convincingly shown. Second, these approaches assume that the camera poses are known precisely for all provided observations. This is a practically and biologically unrealistic assumption; an agent typically only has access to its own observations, not its precise location relative to objects in the scene (albeit it is provided by some oracle in synthetic environments, *e.g.*, [6]). Third, there are no constraints to guarantee consistency among the synthesized results.

In this paper, we address these issues with a unified framework that incrementally generates complete 2D or 3D scenes (*c.f.* Figure 1). Our solution builds upon the MapNet system [11], which offers an elegant solution to the registration problem but has no memory-reading capability. In comparison, our method not only provides a completely functional memory system, but also displays superior generation performance when compared to parallel deep reinforcement learning methods (*e.g.*, [8]). To the best of our knowledge, our solution is the first complete end-to-end trainable read/write allocentric spatial memory for visual inputs. Our key contributions are summarized below:

- Starting with only scene observations from a non-localized agent (*i.e.*, no location/action inputs unlike, *e.g.*, [8]), we present novel mechanisms to update a global memory with encoded features, hallucinate unobserved regions and query the memory for novel view synthesis.

- Memory updates are done with either observed or hallucinated data. Our domain-aware mechanism is the first to explicitly ensure the representation's global consistency w.r.t. the underlying scene properties in both cases.

- We propose the first framework that integrates observation, localization, globally consistent scene learning, and hallucination-aware representation updating to enable incremental scene synthesis.

We demonstrate the efficacy of our framework on a variety of partially observable synthetic and realistic 2D environments. Finally, to establish scalability, we also evaluate the proposed model on challenging 3D environments.

## 2  Related Work

Our work is related to localization, mapping, and novel view synthesis. We discuss relevant work to provide some context.

**Neural Localization and Mapping.** The ability to build a global representation of an environment, by registering frames captured from different viewpoints, is key to several concepts such as reinforcement learning or scene reconstruction. Recurrent neural networks are commonly used to accumulate features from image sequences, *e.g.*, to predict the camera trajectory [15, 19]. Extending these solutions with a queryable memory, state-of-the-art models are mostly egocentric and action-conditioned [3, 17, 30, 8, 14]. Some oracle is, therefore, usually required to provide the agent's action at each time step $t$ [14]. This information is typically used to regress the agent state $s_t$, *e.g.*, its pose,

which can be used in a memory structure to index the corresponding observation $x_t$ or its features. In comparison, our method solely relies on the observations to regress the agent's pose.

Progress has also been made towards solving visual SLAM with neural networks. CNN-SLAM [23] replaced some modules in classical SLAM methods [5] with neural components. Neural SLAM [30] and MapNet [11] both proposed a spatial memory system for autonomous agents. Whereas the former deeply interconnects memory operations with other predictions (*e.g.*, motion planning), the latter offers a more generic solution with no assumption on the agents' range of action or goal. Extending MapNet, our proposed model not only attempts to build a map of the environment, but also makes incremental predictions and hallucinations based on both past experiences and current observations.

**3D Modeling and Geometry-based View Synthesis.** Much effort has also been expended in explicitly modeling the underlying 3D structure of scenes and objects, *e.g.*, [5, 4]. While appealing and accurate results are guaranteed when multiple source images are provided, this line of work is fundamentally not able to deal with sparse inputs. To address this issue, Flynn *et al.* [7] proposed a deep learning approach focused on the multi-view stereo problem by regressing directly to output pixel values. On the other hand, Ji *et al.* [12] explicitly utilized learned dense correspondences to predict the image in the middle view of two source images. Generally, these methods are limited to synthesizing a middle view among fixed source images, whereas our framework is able to generate arbitrary target views by extrapolating from prior domain knowledge.

**Novel View Synthesis.** The problem we tackle here can be formulated as a novel view synthesis task: given pictures taken from certain poses, solutions need to synthesize an image from a new pose, and has seen significant interest in both vision [16, 31] and graphics [10]. There are two main flavors of novel view synthesis methods. The first type synthesizes pixels from an input image and a pose change with an encoder-decoder structure [22]. The second type reuses pixels from an input image with a sampling mechanism. For instance, Zhou *et al.* [31] recasted the task of novel view synthesis as predicting dense flow fields that map the pixels in the source view to the target view, but their method is not able to hallucinate pixels missing from source view. Recently, methods that use geometry information have gained popularity, as they are more robust to large view changes and resulting occlusions [16]. However, these conditional generative models rely on additional data to perform their target tasks. In contrast, our proposed model enables the agent to predict its own pose and synthesize novel views in an end-to-end fashion.

## 3    Methodology

While the current state of the art in scene registration yields satisfying results, there are several assumptions, including prior knowledge of the agent's range of actions, as well as the actions $a_t$ themselves at each time step. In this paper, we consider unknown agents, with only their observations $x_t$ provided during the memorization phase. In the spirit of the MapNet solution [11], we use an allocentric spatial memory map. Projected features from the input observations are registered together in a coordinate system relative to the first inputs, allowing to regress the position and orientation (*i.e.*, *pose*) of the agent in this coordinate system at each step. Moreover, given viewpoints and camera intrinsic parameters, features can be extracted from the spatial memory (*frustum culling*) to recover views. Crucially, at each step, memory "holes" can be temporarily filled by a network trained to generate domain-relevant features while ensuring global consistency. Put together (*c.f.* Figure 2), our pipeline (trainable both separately and end-to-end) can be seen as an explicit topographic memory system with localization, registration, and retrieval properties, as well as consistent memory-extrapolation from prior knowledge. We present details of our proposed approach in this section.

### 3.1    Localization and Memorization

Our solution first takes a sequence of observed images $x_t \in \mathbb{R}^{c \times h \times w}$ (*e.g.*, with $c = 3$ for RGB images or 4 for RGB-D ones) for $t = 1, \ldots, \tau$ as input, localizing them and updating the spatial memory $m \in \mathbb{R}^{n \times u \times v}$ accordingly. The memory $m$ is a discrete global map of dimensions $u \times v$ and feature size $n$. $m_t$ represents its state at time $t$, after updating $m_{t-1}$ with features from $x_t$.

**Encoding Memories.** Observations are encoded to fit the memory format. For each observation, a feature map $x'_t \in \mathbb{R}^{n \times h' \times w'}$ is extracted by an encoding convolutional neural network (CNN). Each feature map is then projected from the 2D image domain into a tensor $o_t \in \mathbb{R}^{n \times s \times s}$ representing the

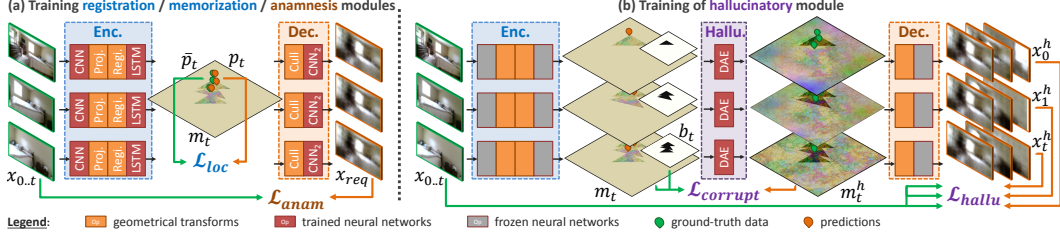

**Figure 3: Pipeline training.** Though steps are shown separately in the figure (for clarity), the method is trained in a single pass. $\mathcal{L}_{loc}$ measures the accuracy of the predicted allocentric poses, *i.e.*, training the encoding system to extract meaningful features (CNN) and to update the global map $m_t$ properly (LSTM). $\mathcal{L}_{anam}$ measures the quality of the images rendered from $m_t$ using the ground-truth poses, to train the decoding CNN. $\mathcal{L}_{hallu}$ trains the method to predict all past and future observations at each step of the sequence, while $\mathcal{L}_{corrupt}$ punishes it for any memory corruption during hallucination.

agent's spatial neighborhood (to simplify later equations, we assume $u, v, s$ are odd). This operation is data and use-case dependent. For instance, for RGB-D observations of 3D scenes (or RGB images extended by some monocular depth estimation method, *e.g.*, [28]), the feature maps are first converted into point clouds using the depth values and the camera intrinsic parameters (assuming like Henriques and Vedaldi [11] that the ground plane is approximately known). They are then projected into $o_t$ through discretization and max-pooling (to handle many-to-one feature aggregation, *i.e.*, when multiple features are projected into the same cell [18]). For 2D scenes (*i.e.*, agents walking on an image plane), $o_t$ can be directly obtained from $x_t$ (with optional cropping/scaling).

**Localizing and Storing Memories.** Given a projected feature map $o_t$ and the current memory state $m_{t-1}$, the registration process involves densely matching $o_t$ with $m_{t-1}$, considering all possible positions and rotations. As explained in Henriques and Vedaldi [11], this can be efficiently done through cross-correlation. Considering a set of $r$ yaw rotations, a bank $o'_t \in \mathbb{R}^{r \times n \times s \times s}$ is built by rotating $o_t$ $r$ times: $o'_t = \left\{ R(o_t , 2\pi\frac{i}{r} , c_{s,s}) \right\}_{i=0}^r$, with $c_{s,s} = (\frac{s+1}{2}, \frac{s+1}{2})$ horizontal center of $o_t$, and $R(o, \alpha, c)$ the function rotating each element in $o$ around the position $c$ by an angle $\alpha$, in the horizontal plane. The dense matching can therefore be achieved by sliding this bank of $r$ feature maps across the global memory $m_{t-1}$ and comparing the correlation responses. The localization probability field $p_t \in \mathbb{R}^{r \times u \times v}$ is efficiently obtained by computing the cross-correlation (*i.e.*, "*convolution*", operator $\star$, in deep learning literature) between $m_{t-1}$ and $o'_t$ and normalizing the response map (*softmax* activation $\sigma$). The higher a value in $p_t$, the stronger the belief the observation comes from the corresponding pose. Given this probability map, it is possible to register $o_t$ into the global map space (*i.e.*, rotating and translating it according to $p_t$ estimation) by directly convolving $o_t$ with $p_t$. This registered feature tensor $\hat{o}_t \in \mathbb{R}^{n \times u \times v}$ can finally be inserted into memory:

$$m_t = \text{LSTM}(m_{t-1}, \hat{o}_t, \theta_{lstm}) \quad \text{with} \quad \hat{o}_t = p_t * o'_t \quad \text{and} \quad p_t = \sigma(m_{t-1} \star o'_t) \tag{1}$$

A long short-term memory (LSTM) unit is used, to update $m_{t-1}$ (the unit's *hidden* state) with $\hat{o}_t$ (the unit's input) in a knowledgeable manner (*c.f.* trainable parameters $\theta_{lstm}$). During training, the recurrent network will indeed learn to properly blend overlapping features, and to use $\hat{o}_t$ to solve potential uncertainties in previous insertions (uncertainties in $p$ result in blurred $\hat{o}$ after convolution). The LSTM is also trained to update an occupancy mask of the global memory, later used for constrained hallucination (*c.f.* Section 3.3).

**Training.** The aforementioned process is trained in a supervised manner given the ground-truth agent's poses. For each sequence, the feature vector $o_{t=0}$ from the first observation is registered at the center of the global map without rotation (origin of the allocentric system). Given $\bar{p}_t$, the one-hot encoding of the actual state at time $t$, the network's loss $\mathcal{L}_{loc}$ at time $\tau$ is computed over the remaining predicted poses using binary cross-entropy:

$$\mathcal{L}_{loc} = -\frac{1}{\tau} \sum_{t=1}^{\tau} \left[ \bar{p}_t \cdot \log(p_t) + (1 - \bar{p}_t) \cdot \log(1 - p_t) \right] \tag{2}$$

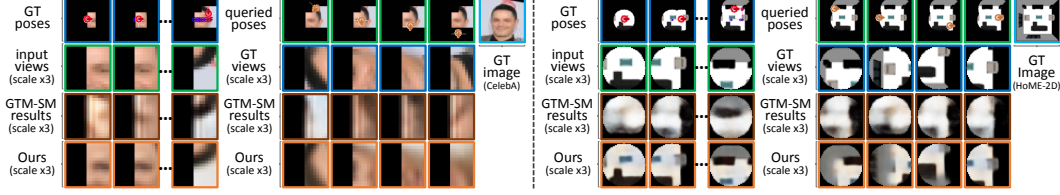

Figure 4: **Synthesis of memorized and novel views** from 2D scenes, comparing to GTM-SM [8]. Methods receive a sequence of 10 observations (along with the related actions for GTM-SM) from an exploring agent, then they apply their knowledge to generate 46 novel views. GTM-SM has difficulties grasping the structure of the environment from short observation sequences, while our method usually succeeds thanks to prior knowledge.

### 3.2 Anamnesis

Applying a novel combination of geometrical transforms and decoding operations, memorized content can be recalled from $m_t$ and new images from unexplored locations synthesized. This process can be seen as a many-to-one recurrent generative network, with image synthesis conditioned on the global memory and the requested viewpoint. We present how the entire network can thus be advantageously trained as an auto-encoder with a recurrent neural encoder and a persistent latent space.

**Culling Memories.** While a decoder can retrieve observations conditioned on the full memory and requested pose, it would have to disentangle the visual and spatial information itself, which is not trivial to learn (*c.f.* ablation study in Section 4.1). Instead, we propose to use the spatial properties of our memory to first *cull* features from requested viewing volumes before passing them as inputs to our decoder. More formally, given the allocentric coordinates $l_{req} = (u_{req}, v_{req})$, orientation $\alpha_{req} = 2\pi \frac{r_{req}}{r}$, and field of view $\alpha_{fov}$, $o_{req} \in \mathbb{R}^{n \times s \times s}$ representing the requested neighborhood is filled as follow:

$$o_{req,kij} = \begin{cases} \hat{o}_{req,kij} & \text{if atan2} \frac{j - \frac{s+1}{2}}{i - \frac{s+1}{2}} < \frac{\alpha_{fov}}{2} \\ -1 & \text{otherwise} \end{cases} \quad (3)$$

with $\hat{o}_{req}$ the unculled feature patch extracted from $m_t$ rotated by $-\alpha_{req}$, *i.e.*, $\forall k \in [0 .. n - 1]$, $\forall (i, j) \in [0 .. s - 1]^2$:

$$\hat{o}_{req,kij} = R(m_t, -\alpha_{req}, c_{u,v} + l_{req})_{k\xi\eta} \quad \text{with} \quad (\xi, \eta) = (i, j) + c_{u,v} + l_{req} - c_{s,s} \quad (4)$$

This differentiable operation combines feature extraction (through translation and rotation) and *viewing frustum culling* (*c.f.* computer graphics to render large 3D scenes).

**Decoding Memories.** As input observations undergo encoding and projection, feature maps culled from the memory go through a reverse procedure to be projected back into the image domain. With the synthesis conditioning covered in the previous step, a decoder directly takes $o_{req}$ (*i.e.*, the view-encoding features) and returns $x_{req}$, the corresponding image. This back-projection is still a complex task. The decoder must both project the features from voxel domain to image plane, and decode them into visual stimuli. Previous works and qualitative results demonstrate that a well-defined (*e.g.*, *geometry-aware*) network can successfully accomplish this task.

**Training.** By requesting the pipeline to recall given observations—*i.e.*, setting $l_{req,t} = \bar{l}_t$ and $r_{req,t} = \bar{r}_t$, $\forall t \in [1, \tau]$, with $\bar{l}_t$ and $\bar{r}_t$ the agent's ground-truth position/orientation at each step $t$—it can be trained end-to-end as an image-sequence auto-encoder (*c.f.* Figure 3.a). Therefore, its loss $\mathcal{L}_{anam}$ is computed as the L1 distance between $x_t$ and $x_{req,t}$, $\forall t \in [0, \tau]$, averaged over the sequences. Note that thanks to our framework's modularity, the global map and registration steps can be removed to pre-train the encoder and decoder together (passing the features directly from one to the other). We observe that such a pre-training tends to stabilize the overall learning process.

### 3.3 Mnemonic Hallucination

While the presented pipeline can generate novel views, these views have to overlap with previous observations for the solution to extract enough features for anamnesis. Therefore, we extend our memory system with an *extrapolation* module to *hallucinate* relevant features for unexplored regions.

**Hole Filling with Global Constraints.** Under global constraints, we build a deep auto-encoder (DAE) in the feature domain, which takes $m_t$ as input, as well as a noise vector of variable amplitude (*e.g.*, no noise for deterministic navigation planning or heavy noise for image dataset augmentation), and returns a convincingly hole-filled version $m_t^h$, while leaving registered features uncorrupted. In other words, this module should provide relevant features while seamlessly integrating existing content according to prior domain knowledge.

**Training.** Assuming the agent homogeneously explores training environments, the hallucinatory module is trained at each step $t \in [0, \tau - 1]$ by generating $m_t^h$, the hole-filled memory used to predict yet-to-be-observed views $\{x_i\}_{i=t+1}^{\tau}$. To ensure that registered features are not corrupted, we also verify that all observations $\{x_i\}_{i=0}^{t}$ can be retrieved from $m_t^h$ (*c.f.* Figure 3.b). This generative loss is computed as follows:

$$\mathcal{L}_{hallu} = \frac{1}{\tau(\tau - 1)} \sum_{t=0}^{\tau-1} \sum_{i=0}^{\tau} |x_{i,t}^h - x_i|_1 \tag{5}$$

with $x_{i,t}^h$ the view recovered from $m_t^h$ using the agent's true location $\bar{l}_i$ and orientation $\bar{r}_i$ for its observation $x_i$. Additionally, another loss is directly computed in the feature domain, using memory occupancy masks $b_t$ to penalize any changes to the registered features (given $\odot$ Hadamard product):

$$\mathcal{L}_{corrupt} = \frac{1}{\tau} \sum_{t=0}^{\tau} |(m_t^h - m_t) \odot b_t|_1 \tag{6}$$

Trainable end-to-end, our model efficiently acquires domain knowledge to register, hallucinate, and synthesize scenes.

## 4 Experiments

We demonstrate our solution on various synthetic and real 2D and 3D environments. For each experiment, we consider an unknown agent exploring an environment, only providing a short sequence of partial observations (limited field of view). Our method has to localize and register the observations, and build a global representation of the scene. Given a set of requested viewpoints, it should then render the corresponding views. In this section, we qualitatively and quantitatively evaluate the predicted trajectories and views, comparing with GTM-SM [8], the only other end-to-end memory system for scene synthesis, based on the Generative Query Network [6].

### 4.1 Navigation in 2D Images

We first study agents exploring images (randomly walking, accelerating, rotating), observing the image patch in their field of view at each step (more details and results in the supplementary material).

**Experimental Setup.** We use a synthetic dataset of indoor $83 \times 83$ floor plans rendered using the HoME platform [2] and SUNCG data [20] (8,640 training + 2,240 test images from random rooms "*office*", "*living*", and "*bedroom*"). Similar to Fraccaro *et al.* [8], we also consider an agent exploring real pictures from the CelebA dataset [13], scaled to $43 \times 43$px. We consider two types of agents for each dataset. To reproduce Fraccaro *et al.* [8] experiments, we first consider non-rotating agents $A^s$—only able to translate in the 4 directions—with a $360°$ field of view covering an image patch centered on the agents' position. The CelebA agent $A_{cel}^s$ has a $15 \times 15$px square field of view; while the field of view of the HoME-2D agent $A_{hom}^s$ reaches 20px away, and is therefore circular (in the $41 \times 41$ patches, pixels further than 20px are left blank). To consider more complex scenarios, agents $A_{cel}^c$ and $A_{hom}^c$ are also designed. They can rotate and translate (in the gaze direction), observing patches rotated accordingly. On CelebA images, $A_{cel}^c$ can rotate by $\pm 45°$ or $\pm 90°$ each step, and only observes $8 \times 15$ patches in front ($180°$ rectangular field of view); while for HoME-2D, $A_{hom}^c$ can rotate by $\pm 90°$ and has a $150°$ field of view limited to 20px. All agents can move from $1/4$ to $3/4$ of their field of view each step. Input sequences are 10 steps long. For quantitative studies, methods have to render views covering the whole scenes w.r.t. the agents' properties.

**Qualitative Results.** As shown in Figure 4, our method efficiently uses prior knowledge to register observations and extrapolate new views, consistent with the global scene and requested viewpoints. While an encoding of the agent's actions is also provided to GTM-SM (guiding the localization), it

Table 1: **Quantitative comparison on 2D and 3D scenes**, *c.f*. setups in Subsections 4.1-4.2 ($\searrow$ *the lower the better;* $\nearrow$ *the higher the better; "u" horizontal bin unit according to AVD setup*).

| Exp. | Methods | Average Position Error | | | Absolute Trajectory Error | | | Anam. Metr. | | Hall. Metr. | |
|---|---|---|---|---|---|---|---|---|---|---|---|
| | | Med.$\searrow$ | Mean$\searrow$ | Std.$\searrow$ | Med.$\searrow$ | Mean$\searrow$ | Std.$\searrow$ | L1$\searrow$ | SSIM$\nearrow$ | L1$\searrow$ | SSIM$\nearrow$ |
| A) $A_{cel}^s$ | GTM-SM | 4.0px | 4.78px | 4.32px | 6.40px | 6.86px | 3.55px | 0.14 | 0.57 | 0.14 | 0.41 |
| | GTM-SM$^{L1_{s_t \leftrightarrow l_t}}$ * | 1.0px | 1.03px | 1.23px | 0.79px | 0.87px | 0.86px | 0.13 | 0.64 | 0.15 | 0.40 |
| | GTM-SM$^{s_t \leftarrow l_t}$ ** | 0px (NA – poses passed as inputs) | | | 0px (NA – poses passed as inputs) | | | 0.08 | 0.76 | 0.13 | 0.43 |
| | Ours | **1.0px** | **0.68px** | **1.02px** | **0.49px** | **0.60px** | **0.64px** | **0.06** | **0.80** | **0.09** | **0.72** |
| B) $A_{cel}^c$ | GTM-SM | 3.60px | 5.04px | 4.42px | 2.74px | 1.97px | 2.48px | 0.21 | 0.50 | 0.32 | 0.41 |
| | Ours | **1.0px** | **2.21px** | **3.76px** | **1.44px** | **1.72px** | **2.25px** | **0.08** | **0.79** | **0.20** | **0.70** |
| C) $A_{hom}^s$ | GTM-SM | 4.0px | 4.78px | 4.32px | 6.40px | 6.86px | 3.55px | 0.14 | 0.57 | 0.14 | 0.41 |
| | Ours | **1.0px** | **0.68px** | **1.02px** | **0.49px** | **0.60px** | **0.64px** | **0.06** | **0.80** | **0.09** | **0.72** |
| D) Doom | GTM-SM | 1.41u | 2.15u | **1.84u** | **1.73u** | **1.81u** | **1.06u** | 0.09 | 0.52 | 0.13 | 0.49 |
| | Ours | **1.00u** | **1.64u** | 2.16u | 1.75u | 1.95u | 1.24u | 0.09 | **0.56** | **0.11** | **0.54** |
| E) AVD | GTM-SM | 1.00u | 0.77u | 0.69u | 0.31u | 0.36u | 0.40u | 0.37 | 0.12 | 0.43 | 0.10 |
| | Ours | **0.37u** | **0.32u** | **0.26u** | **0.20u** | **0.21u** | **0.18u** | **0.22** | **0.31** | **0.25** | **0.23** |

\* GTM-SM$^{L1_{s_t \leftrightarrow l_t}}$: Custom GTM-SM with a L1 localization loss computed between the predicted states $s_t$ and ground-truth poses $l_t$.
\*\* GTM-SM$^{s_t \leftarrow l_t}$: Custom GTM-SM with the ground-truth poses $l_t$ provided as input (no $s_t$ inference).

Table 2: **Ablation study** on CelebA with agent $A_{cel}^c$. Removed modules are replaced by identity mappings; remaining ones are adapted to the new input shapes when necessary. LSTM, memory, and decoder are present in all instances ("Localization" is the MapNet module).

| Pipeline Modules | | | | Anamnesis Metrics | | Hallucination Metrics | |
|---|---|---|---|---|---|---|---|
| Encoder | Localization | Hallucinatory DAE | Culling | L1$\searrow$ | SSIM$\nearrow$ | L1$\searrow$ | SSIM$\nearrow$ |
| ∅ | ∅ | ∅ | ∅ | 0.18 | 0.62 | 0.24 | 0.59 |
| ✓ | ∅ | ∅ | ∅ | 0.17 | 0.62 | 0.24 | 0.58 |
| ✓ | ✓ | ∅ | ∅ | 0.15 | 0.66 | 0.20 | 0.61 |
| ✓ | ✓ | ✓ | ∅ | 0.15 | 0.65 | 0.19 | 0.62 |
| ✓ | ∅ | ✓ | ✓ | 0.14 | 0.69 | 0.19 | 0.63 |
| ∅ | ✓ | ✓ | ✓ | 0.13 | 0.71 | 0.17 | 0.66 |
| ✓ | ✓ | ∅ | ✓ | **0.08** | **0.80** | 0.18 | 0.66 |
| ✓ | ✓ | ✓ | ✓ | **0.08** | **0.80** | **0.15** | **0.70** |

cannot properly build a global representation from short input sequences, and thus fails at rendering completely novel views. Moreover, unlike the dictionary-like memory structure of GTM-SM, our method stores its representation into a single feature map, which can therefore be queried in several ways. As shown in Figure 6, for a varying number of conditioning inputs, one can request novel views one by one, culling and decoding features; with the option to register hallucinated views back into memory (*i.e.*, saving them as "valid" observations to be reused). But one can also directly query the full memory, training another decoder to convert all the features. Figure 6 also demonstrates how different trajectories may lead to different intermediate representations, while Figure 7-a illustrates how the proposed model can predict different global properties for identical trajectories but different hallucinatory noise. In both cases though (different trajectories or different noise), the scene representations converge as the scene coverage increases.

**Quantitative Evaluations.** We quantitatively evaluate the methods' ability to register observations at the proper positions in their respective coordinate systems (*i.e.*, to predict agent trajectories), to retrieve observations from memory, and to synthesize new ones. For localization, we measure the average position error (APE) and the absolute trajectory error (ATE), commonly used to evaluate SLAM systems [4].

For image synthesis, we make the distinction between recalling images already observed (*anamnesis*) and generating unseen views (*hallucination*). For both, we compute the common L1 distance between predicted and expected values, and the structural similarity (SSIM) index [25] for the assessment of perceptual quality [24, 29].

Table 1.A-C shows the comparison on 2D cases. For pose estimation, our method is generally more precise even though it leverages only the observations to infer trajectories, whereas GTM-SM also infers more directly from the provided agent actions. However, GTM-SM is trained in an unsupervised manner, without any location information. Therefore, we extend our evaluation by comparing our method with two custom GTM-SM solutions that leverage ground-truth poses during training (supervised L1 loss over the predicted states/poses) and inference (poses directly provided

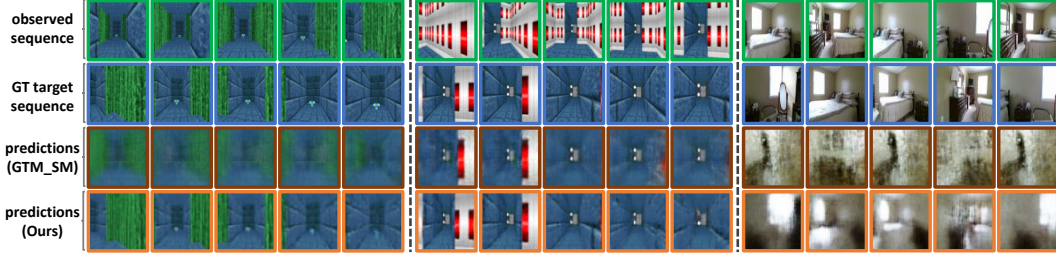

Figure 5: **Qualitative comparison on 3D use-cases**, w.r.t. anamnesis and hallucination.

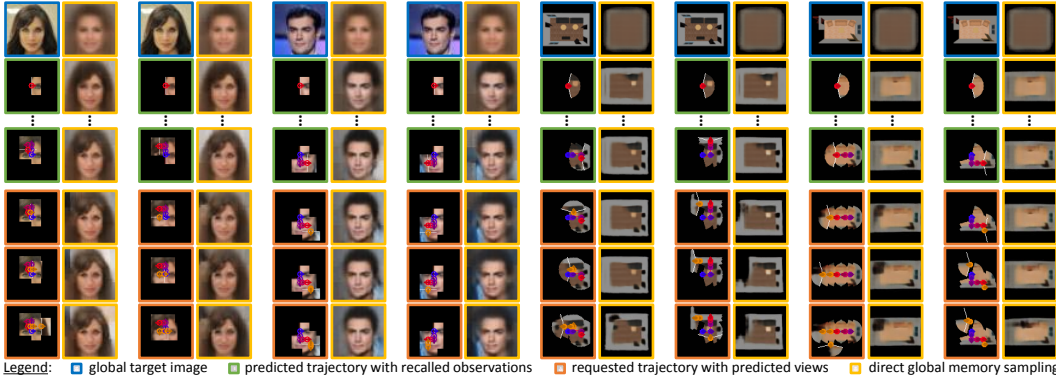

Legend: □ global target image  □ predicted trajectory with recalled observations  □ requested trajectory with predicted views  □ direct global memory sampling

Figure 6: **Incremental exploration and hallucination** (on 2D data). Scene representations evolve with the registration of observed or hallucinated views (*e.g.*, adapting hair color, face orientation,*etc.*).

as additional inputs). While these changes unsurprisingly improve the accuracy of GTM-SM, our method is still on a par with these results (*c.f.* Table 1.A).

Moreover, while GTM-SM fares well enough in recovering seen images from memory, it cannot synthesize views out of the observed domain. Our method not only extrapolates adequately from prior knowledge, but also generates views which are consistent from one to another (*c.f.* Figure 6 showing views stitched into a consistent global image). Moreover, as the number of observations increases, so does the quality of the generated images (*c.f.* Figure 7-b), Note that on a Nvidia Titan X, the whole process (registering 5 views, localizing the agent, recalling the 5 images, and generating 5 new ones) takes less than 1s.

**Ablation Study.** Results of an ablation study are shown in Table 2 to further demonstrate the contribution of each module. Note that the APE/ATE are not represented, as they stay constant as long as the MapNet localization is included. In other words, our extensions cause no regression in terms of localization. Localizing and clipping features facilitate the decoding process by disentangling the visual and spatial information, thus improving the synthesis quality. Hallucinating features directly in the memory ensures image consistency.

## 4.2 Exploring Virtual and Real 3D Scenes

We finally demonstrate the capability of our method on the more complex case of 3D scenes.

**Experimental Setup.** As a first 3D experiment, we recorded, with the Vizdoom platform [27], 34 training and 6 testing episodes of 300 RGB-D observations from a human-controlled agent navigating in various static virtual scenes (walking with variable speed or rotating by $30°$ each step). Poses are discretized into 2D bins of $30 \times 30$ game units. Trajectories of 10 continuous frames are sampled and passed to the methods (the first 5 images as observations, and the last 5 as training ground-truths). We then consider the Active Vision Dataset (AVD) [1] which covers various real indoor scenes, often capturing several rooms per scene. We selected $15$ for training and $4$ for testing as suggested by the dataset authors, for a total of $\sim 20,000$ RGB-D images densely captured every 30cm (on a 2D grid) and every $30°$ in rotation. For each scene we randomly sampled $5,000$ agent trajectories of 10 frames each (each step the agent goes forward with $70\%$ probability or rotates either way, to

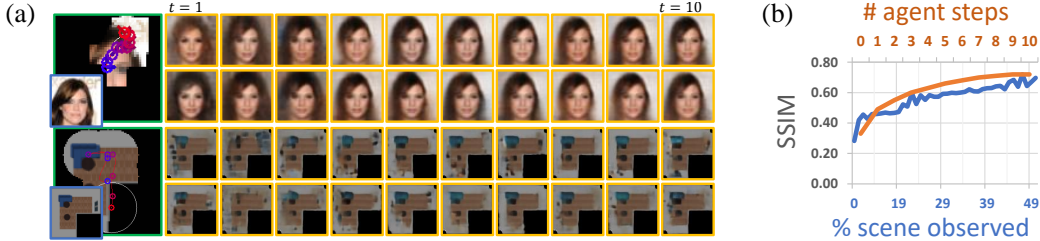

Figure 7: (a) **Statistical nature of the hallucinated content**. Global scene representations are shown for each step $t$, given the same agent trajectories but different noise vectors passed to the hallucinatory auto-encoder; (b) **Salient image quality w.r.t. agent steps and scene coverage** for $A^s_{cel}$, computed over the global scene representations. These results show how the global scene properties converge and the quality of generated images increase as observations accumulate.

favor exploration). For both experiments, the 10-frame sequences are passed to the methods—the first 5 images as observations and the last 5 as ground-truths during training. Again, GTM-SM also receives the action encodings. For our method, we opted for $m \in \mathbb{R}^{32 \times 43 \times 43}$ for the Doom setup and $m \in \mathbb{R}^{32 \times 29 \times 29}$ for the AVD one.

**Qualitative Results.** Though a denser memory could be used for more refined results, Figure 5 shows that our solution is able to register meaningful features and to understand scene topographies simply from 5 partial observations. We note that quantization in our method is an application-specific design choice rather than a limitation. When compute power and memory allow, finer quantization can be used to obtain better localization accuracy (*c.f.* comparisons and discussion presented by MapNet authors [11]). In our case, relatively coarse quantization is sufficient for scene synthesis, where the global scene representation is more crucial. In comparison, GTM-SM generally fails to adapt the VAE prior and predict the belief of target sequences (refer to the supplementary material for further results).

**Quantitative Evaluation.** Adopting the same metrics as in Section 4.1, we compare the methods. As seen in Table 1.D-E, our method slightly underperforms in terms of localization in the Doom environment. This may be due to the approximate rendering process VizDoom uses for the depth observations, with discretized values not matching the game units. Unlike GTM-SM which relies on action encodings for localization, these unit discrepancies affect our observation-based method. As to the quality of retrieved and hallucinated images, our method shows superior performance (*c.f.* additional saliency metrics in the supplementary material). While current results are still far from being visually pleasing, the proposed method is promising, with improvements expected from more powerful generative networks.

It should also be noted that the proposed hallucinatory module is more reliable when target scenes have learnable priors (*e.g.*, structure of faces). Hallucination of uncertain content (*e.g.*, layout of a 3D room) can be of lower quality due to the trade-off between representing uncertainties w.r.t. missing content and unsure localization, and synthesizing detailed (but likely incorrect) images. Soft registration and hallucinations' statistical nature can add "uncertainty" leading to blurred results, which our generative components partially compensate for (*c.f.* our choice of a GAN solution for the DAE to improve its sampling, *c.f.* supplementary material). For data generation use-cases, relaxing hallucination constraints and scaling up $\mathcal{L}_{hallu}$ and $\mathcal{L}_{anam}$ can improve image detail at the price of possible memory corruption (we focused on consistency rather than high-resolution hallucinations).

## 5 Conclusion

Given unlocalized agents only providing observations, our framework builds global representations consistent with the underlying scene properties. Applying prior domain knowledge to harmoniously complete sparse memory, our method can incrementally sample novel views over whole scenes, resulting in the first complete read and write spatial memory for visual imagery. We evaluated on synthetic and real 2D and 3D data, demonstrating the efficacy of the proposed method's memory map. Future work can involve densifying the memory structure and borrowing recent advances in generating high-quality images with GANs [26].

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
