[Supplementary Material]

# Incremental Scene Synthesis – Supplementary Material

**Benjamin Planche**[1,2]    **Xuejian Rong**[3,4]    **Ziyan Wu**[4]    **Srikrishna Karanam**[4]
**Harald Kosch**[2]    **YingLi Tian**[3]    **Jan Ernst**[4]    **Andreas Hutter**[1]

[1]Siemens Corporate Technology, Munich, Germany
[2]University of Passau, Passau, Germany
[3]The City College, City University of New York, New York NY
[4]Siemens Corporate Technology, Princeton NJ
{first.last}@siemens.com, {xrong,ytian}@ccny.cuny.edu, harald.kosch@uni-passau.de

In the following sections, we introduce further pipeline details for reproducibility. We also provide various additional qualitative results (Figures S3 to S7) and quantitative comparisons (Section 2.3) on 2D and 3D datasets. A video is also attached, presenting our solution applied to the incremental registration of unlocalized observations and generation of novel views.

## 1 Methodology and Implementation Details

This section contains further details regarding the several interlaced components of our pipeline and their implementation.

### 1.1 Localization and Memorization

#### 1.1.1 Encoding Memories

Observations are encoded using a shallow ResNet [8] with 4 residual blocks (reusing the *CycleGAN* custom implementation of zhis architecture [28]). The encoder $E$ is thus configured to output feature maps $x'_t \in \mathbb{R}^{n \times h' \times w'}$ with the same dimensions as the inputs $x_t \in \mathbb{R}^{c \times h \times w}$, *i.e.* $h = h', w = w'$.

As explained in Section 3.1, the projection of $x'_t \in \mathbb{R}^{n \times h' \times w'}$ (with features in the image coordinate system) into $o_t \in \mathbb{R}^{n \times s \times s}$, the representation of the agent's spatial neighborhood, is use-case dependent. For 2D image exploration, this operation is done by cropping $x'_t$ into a square tensor $n \times s' \times s'$ with $s' = min(h', w')$, followed by scaling the features from $s' \times s'$ to $s \times s$ using bilinear interpolation.

For 3D use-cases with RGB-D observations, the input depth maps $x^d_t$ are used to project $x'_t$ into a 3D point cloud (after registering color and depth images together), before converting this sparse representation into a dense tensor using max-pooling. For the 3D projection, $\forall i \in \{0, ..., h-1\}$ and $\forall j \in \{0, ..., w-1\}$, each feature $x_{t,i,j} \in \mathbb{R}^n$ of $x'_t$ receives the coordinates $(x, y, z)$ similar to [9]:

$$z = x^d_{t,i,j} \quad ; \quad x = (j - c_x)\frac{z}{f_x} \quad ; \quad y = (i - c_y)\frac{z}{f_y} \tag{1}$$

with $f_x, f_y$ the pixel focal lengths of the depth sensor, and $c_x, c_y$ its pixel focal center (for KinectV2: $f_x = 366.193$px, $f_y = 365.456$px, $c_x = 256.684$px, $c_y = 207.085$px for $512 \times 424$ images).

Each set of coordinates is then discretized to obtain the neighborhood bin the feature belongs to. Given $s \times s$ bins of dimensions $(x_s, z_s)$ in world units, the bin coordinates $(x_b, z_b)$ of each feature are computed as follow:

$$x_b = \lfloor \frac{x}{x_s} \rfloor + \frac{s-1}{2} \quad ; \quad z_b = \lfloor \frac{z}{z_s} \rfloor + \frac{s-1}{2} \tag{2}$$

Figure S1: **Localization and memorization**, based on MapNet [9].

with $\lfloor \cdot \rfloor$ the integer flooring operation. Features projected out of the $s \times s$ area are ignored.

Finally, $o_t$ is obtained by applying a max-pooling operation over each bin (*i.e.*, keeping only the maximum values for features projected into the same bin). Empty bins result in a null value[1].

### 1.1.2 Localizing and Storing Memories

Once the features are localized and registered into the allocentric system (*c.f.* Figure S1), an LSTM is used to update the global memory accordingly, *c.f.* Section 3.1. Following the original MapNet solution [9], each spatial location is updated independently to preserve spatial invariance, sharing weights between each LSTM cell. The occupancy mask $b_t$ of the global memory is updated in a similar manner, using an LSTM with shared-weights to update the memory mask with a binary version of $o_t$ (*i.e.* 1 for bins containing projected features, 0 otherwise).

## 1.2 Anamnesis

### 1.2.1 Memory Culling

Figure S2 illustrates the geometrical process to extract features from the global map corresponding to the requested viewpoints and agent's view field angle, as described in Section 3.2. Note that for this step, one can use a larger view field than requested, in order to provide the feature decoder with more context (*e.g.*, to properly recover visual elements at the limit of the agent's view field).

### 1.2.2 Memory Decoding

Similar to the encoder, we use a ResNet-4 architecture [8] for the decoder network, with the last convolutional layers parametrized to output the image tensors $x_{req} \in \mathbb{R}^{c \times h \times w}$ while the network receives inputs feature tensors $o_{req} \in \mathbb{R}^{n \times s \times s}$.

For the experiments where the global memory is directly sampled into an image , another ResNet-4 decoder is trained, directly receiving $m_t^h \in \mathbb{R}^{n \times u \times v}$ for input, returning $x_t^m \in \mathbb{R}^{c \times H \times W}$, and comparing the generated image with the original global image (L1 loss).

## 1.3 Mnemonic Hallucination

For simplicity and homogeneity[2], another network based on the ResNet-4 architecture [8, 28] is used to fill the memory holes. In addition to the global memory $m_t$, the network is adapted to also

Figure S2: **Geometrical Memory Culling** (2D representation). The feature vector $o_{req}$, defining the observation for an agent positioned at $l_{req}$ and rotated by an angle $\alpha_{req}$ in the allocentric system, is extracted from $m_t$ through a series of geometrical transforms (rotation, translation, clipping, culling). Coordinates and distances in the allocentric coordinate system are represented in white; in yellow for the $m_t$ matrix system; and red for the $o_t$ one.

receive as input a noise vector (which can be seen as a random seed for the hallucinations). This vector goes through two fully-connected layers to upsample it, while the main input goes through the downsampling layers of the ResNet; then the two resulting tensors are concatenated before the first residual block.

In order to improve the sampling of hallucinated features and the global awareness of this generator, we adopt several concepts from SAGAN [27]. The ResNet generator is therefore edited as follow:

- Spectral normalization [14] is applied to the weights of each convolution layer in the residual blocks (as SAGAN authors demonstrated it can prevent unusual gradients and stabilize training);

- To model relationships between distant regions, self-attention layers [27, 4, 15, 22] replace the two last convolutions of the network.

Given a feature map $o \in \mathbb{R}^{n \times u \times v}$, the result $o_{sa}$ of the self-attention operation is:

$$o_{sa} = o + \gamma(W_h \star o)\sigma\big((W_f \star o)^{\mathsf{T}}(W_g \star o)\big)^{\mathsf{T}} \tag{3}$$

with $W_f \in \mathbb{R}^{\bar{n} \times n}$, $W_g \in \mathbb{R}^{\bar{n} \times n}$, $W_h \in \ltimes^{n \times n}$ learned weight matrices (we opt for $\bar{n} = {}^n\!/_8$ as in [27]); and $\gamma$ a trainable scalar weight.

Following the generative adversarial network (GAN) strategy [6, 19, 20, 10], our conditioned generator is also trained against a discriminator evaluating the *realism* of feature patches $o_t^h$ culled from $m_t^h$. This discriminator is itself trained against $o_t$ (*real* samples) and $o_t^h$ (*fake* ones). For this network, we also opt for the architecture suggested by Zhang *et al.* [27], *i.e.* a simple convolutional architecture with spectral normalization and self-attention layers.

Given this setup, the generative loss $\mathcal{L}_{hallu}$ is combined to $\mathcal{L}_{disc}$, a discriminative loss obtained by playing the generator $H$ against its discriminator $D$. As a conditional GAN with recurrent elements, the objective this module has to maximize over a complete training sequence is therefore:

$$H^* = \arg\min_H \max_D \mathcal{L}_{disc} + \mathcal{L}_{hallu} + \mathcal{L}_{corrupt} \tag{4}$$

$$\text{with } \mathcal{L}_{disc} = \sum_{t=0}^{\tau} \big[\log D(x_t)\big] + \big[\log\big(1 - D\big(x_t^h\big)\big)\big] \tag{5}$$

### 1.4 Further Implementation Details

Our solution is implemented using the PyTorch framework [16].

**Layer parameterization:**

- Instance normalization is applied inside the ResNet networks;
- All Dropout layers have a dropout rate of $50\%$;
- All LeakyReLU layers have a leakiness of $0.2$.
- Image values are normalized between -1 and 1.

**Training parameters:**

- Weights are initialized from a zero-centered Gaussian distribution, with a standard deviation of $0.02$ ;
- The Adam optimizer [12] is used, with $\beta_1 = 0.5$;
- The base learning rate is initialized at $2e \times 10^{-4}$;
- Training sequence applied in this paper:
    1. Feature encoder and decoder networks are pre-trained together for 10,000 iterations;
    2. The complete memorization and anamnesis process (encoder, LSTM, decoder) is then trained for 10,000 more iterations;
    3. The hallucinatory GAN is then added and the complete solution is trained until convergence.

## 2 Experiments and Results

Additional results are presented in this section. We also provide supplementary information regarding the various experiments we conducted, for reproducibility.

### 2.1 Protocol for Experiments

#### 2.1.1 Comparative Setup

To the best of our knowledge, no other neural method covers agent localization, topographic memorization, scene understanding and relevant novel view synthesis in an end-to-end, integrated manner. The closest state-of-the-art solution to compare with is the recent GTM-SM project [5]. This method uses the differentiable neural dictionary (DND) proposed by Pritzel *et al.* [18] to store encoded observations with the predicted agent's positions for keys. To synthesize a novel view, the $k$-nearest entries (in terms of positions-keys) are retrieved to interpolate the image features, before passing it to a decoder network.

Unlike our method which localizes and registers together the views with no further context needed, GTM-SM requires an encoding of the agent's actions, leading to each new observation, as additional inputs. We thus adapt our data preparation pipeline for this method, so that the agent returns its actions (encoding the direction changes and step lengths) along the observations. At each time step, GTM-SM uses the provided action $a_t$ to regress the agent's state $s_t$ *i.e.* its relative pose in our experiments.

For a fair comparison with the ground-truth trajectories, we thus convert the relative pose sequences predicted by GTM-SM into world coordinates. For that, we apply a least-square optimization process to fit its predicted trajectories over the ground-truth ones *i.e.* computing the most favorable transform to apply before comparison (scaling, rotating and translating the trajectories). For our method, the allocentric coordinates are also converted in world units by scaling the values according to the bin dimensions $(x_s, z_s)$ and applying an offset corresponding to the absolute initial pose of the agent.

#### 2.1.2 Metrics for Quantitative Evaluations

As a reminder (*c.f.* main paper), the following metrics are applied, to evaluate the quality of the localization, the anamnesis, and the hallucination:

global target image | predicted trajectory with registered observations | requested views for full sampling | global memory sampling

Figure S3: **Incremental and direct memory sampling** of complete environments from partial observations (on CelebA).

- The *average position error (APE)* computes the mean Euclidean distance between the predicted positions and their ground-truths for each sequence;

- The *absolute trajectory error (ATE)* is obtained by calculating the root-mean-squared error in the positions of each sequence, after transforming the predicted trajectory to best fit the ground-truth (giving an advantage to GTM-SM predictions through post-processing, as explained in Section 2.1);

- The common *L1 distance* is computed as the per-pixel absolute difference between the predicted and expected values, averaged over each image (recalled and/or hallucinated);

- The *structural similarity (*SSIM*) index* [24, 25], prevalent in the assessment of perceptual quality [23, 26, 7], is computed over $N \times N$ windows extracted from the predicted and ground-truth images, as follow:

$$\text{SSIM}(x, \bar{x}) = \frac{(2\mu_x\mu_{\bar{x}} + c_1) + (2\sigma_{x\bar{x}} + c_2)}{(\mu_x^2 + \mu_{\bar{x}}^2 + c_1)(\sigma_x^2 + \sigma_{\bar{x}}^2 + c_2)} \tag{6}$$

with $x$ and $\bar{x}$ the windows extracted from the predicted and ground-truth images, $\mu_x$ and $\mu_{\bar{x}}$ the mean values of the respective windows, $\sigma_x^2$ and $\sigma_{\bar{x}}^2$ their respective variance, $c_1 = 0.001^2$ and $c_2 = 0.003^2$ two constants for numerical stability. The final index is computed by averaging the values obtained by sliding the windows over the whole images (no overlapping). We opt for $N = 5$ for the CelebA experiments, and $N = 13$ for the HoME-2D ones (*i.e.*, splitting the observations in 9 windows). Note that the closer to 1 the computed index, the better the perceived image quality.

## 2.2 Navigation in 2D Images

### 2.2.1 Experimental Details

The aligned CelebA dataset [13] is split with $197, 599$ real portrait images for training and $5, 000$ for testing. Each image is center-cropped to $160 \times 160$px (in order to remove part of the background and focus on faces), before being scaled to $43 \times 43$px.

We build our synthetic HoME-2D dataset by rendering several thousand RGB floor plans of randomly instantiated rooms using the HoME framework [2] (room categories: "*bedroom*", "*living*", "*office*") and SUNCG data [21]. We use $8, 960$ images for training and $2, 240$ for testing, scaled to $83 \times 83$px.

For both experiments, sequences of observations are generated by randomly walking an agent over the 2D images. At each step, the agent can rotate maximum $\pm 90°$ (for experiments with rotation) and cover a distance from $^1/_4$ to $^3/_4$ its view field radius. However, once a new direction is chosen,

the agent has to take at least 3 steps before being able to rotate again (to favor exploration). The agent is also forced to rotate when one of the image borders is entering its view field.

Each training sequence contains $54$ images for the CelebA experiments, and $41$ for the HoME experiments. Both for GTM-SM [5] and our method, only 10 images are passed as observations to fill and train the topographic memory systems (using the provided ground-truth positions/orientations). The remaining 44 or 31 images (sampled by forcing the agent to follow a pre-determined trajectory covering the complete 2D environments) are used as ground-truth information for the hallucinatory modules of the two pipelines.

As described in Section 4.1, for each dataset we consider two different types of agents, *i.e.* with more or less realistic characteristics:

**Simple agent.** We first consider a non-rotating agent—only able to translate in the four directions— with a $360°$ view field covering an image patch centered on the agent's position. For CelebA experiments, this view field is $15 \times 15$px square patch; while for HoME-2D experiments, the view field reaches 20px away from the agent, and is therefore in the shape of a circular sector (pixels in the corresponding $41 \times 41$ patches further than 20px are set to the null value).

**Advanced agent.** A more realistic agent is also designed, able to rotate and to translate accordingly (*i.e.* in the gaze direction) at each step, observing the image patch in front of it (rotated accordingly). For CelebA experiments, the agent can rotate by $\pm 45°$ or $\pm 90°$ each step, and only observes the $8 \times 15$ patches in front ($180°$ rectangular view field); while for HoME-2D experiments, it can rotate by $\pm 90°$ each step, and has a $150°$ view field limited to 20px.

The first simple agent is defined to reproduce the 2D experiments showcasing GTM-SM [5]. While its authors present some qualitative evaluation with a rotating agent, we were not able to fully reproduce their results, despite the implementation changes we made to take into account the prior dynamics of the moving agent (*i.e.* extending the GTM-SM state space to 3 dimensions; the new third component of the state vectors $s_{t-1}$ is storing the information to build a 2D rotation matrix, itself used with the translation elements to compute $s_t$). We adopt the more realistic agent to demonstrate the capability of our own solution, given its more complex range of actions and partial observations. Rotational errors with this agent are ignored for GTM-SM.

### 2.2.2 Additional Qualitative Results

As explained in the paper, our topographic memory module not only allows to directly build a global representation of the environments, but it also brings the possibility to use prior knowledge to extrapolate the scene content for the unexplored area. In contrast, GTM-SM stores each observation separately in its DND memory [18], and can only generate new views by interpolating between a subset of these entries with a VAE prior. This is illustrated in Figure 3 (comparing the methods on image retrieval from memory and on novel view synthesis) and Figure 4 (showcasing the ability of our pipeline to synthesize complete environments from partial views) in the main paper, as well as in similar Figures S3 to S6.

### 2.3 Exploring Real 3D Scenes

### 2.3.1 Additional Quantitative Results

Additional evaluations were conducted on real 3D data, using the Active Vision Dataset (AVD) [1], in order to demonstrate the salient properties of the generated images (despite their lower visual quality).

First, the Wasserstein metric was computed between the Histogram of Oriented Gradients (HOG) descriptors extracted from the unseen ground-truth images and the corresponding predictions. GTM-SM [5] scored 1.1, whereas our method obtained 0.8 (the lower the better).

Second, we compared the saliency maps of ground-truth and predicted images [3], computing the area-under-the-curve metric (AUC) proposed by Judd *et al.* [11] and the Normalized Scanpath Saliency (NSS) [17]. GTM-SM [5] scored 0.40 for the AUC-Judd and 0.14 for the NSS, whereas our framework obtained 0.63 and 0.38 respectively (the higher the better for both metrics).

### 2.3.2 Additional Qualitative Results

For qualitative comparison with GTM-SM, we trained both methods on AVD dataset with the same setup. Challenges arise from the fact that the 3D environments are much more complex than their 2D counterparts, and more factors need to be considered in memorization and prediction.

Further qualitative results on the AVD test scenes are demonstrated in Figure S7. Given the same observation sequences (additional actions to GTM-SM) and requested poses, the predicted novel views are shown for comparison. Generally, GTM-SM fails to adapt the VAE prior and predict the belief of target sequences, while our method tends to successfully synthesize the room layout based on the learned scene prior and observed images.

Figure S4: **Qualitative comparison with GTM-SM** on CelebA and HoME-2D, in terms of pose / trajectory estimations and in terms of view generation (recovery of seen images from memory and novel view hallucination). Methods receive a sequence of 10 observations (along with the related actions for GTM-SM) from a non-rotating agent exploring the $83 \times 83$ 2D image with a $360°$ circular view field of 20px radius. The methods then apply their knowledge to generate novel views.

Figure S5: **Qualitative results on CelebA dataset**. with sequences of 10 observations from an agent able to rotate and translate every step, exploring the $43 \times 43$ 2D image with a $180°$ view field of $8 \times 15$px (*i.e.* observing the image patch in front of it, rotated accordingly). After each step, the hallucinated features are adapted to blend with the new observations, until reaching convergence as the coverage increases.

Figure S6: **Qualitative results on HoME-2D**. with sequences of 10 observations from an agent able to rotate and translate every step, exploring the $83 \times 83$ 2D image with a $150°$ view field of 20px radius. After each step, the hallucinated features are adapted to blend with the new observations, until reaching convergence as the coverage increases.

Figure S7: **Qualitative comparison with GTM-SM on AVD dataset**, in terms of pose / trajectory estimations and in terms of view generation (recovery of seen images from memory and novel view hallucination). Methods receive a sequence of 5 observations (along with the corresponding actions for GTM-SM) from an agent exploring the testing unseen scenes. The methods then apply their knowledge to generate novel views.

## Footnotes

[1]Max-pooling over a sparse tensor (point cloud) as done here is a complex operation not yet covered by all deep learning frameworks at the time of this project. We thus implemented our own (for the PyTorch library [16]).

[2]Our solution is orthogonal to the choice of encoding/decoding networks. More advanced architectures could be considered.