[Reviews · NeurIPS 2019]

Reviewer 1



Reading the rebuttal and the promised improvements to the writing have increased my score to a 7. ---------------------- The paper presents a spatially-structured memory model capable of registering observations onto a globally consistent map, localizing incoming data and hallucinating areas of the map not yet or partially visited. Although borrowing architectural details from previous work especially with respect to MapNet, the paper proposes a way to incorporate a generative process directly into a spatially structured memory. Previous generative models for scenes have omitted any spatial inductive bias, and present the model directly with the sequence of observations. Additionally, previous spatial architectures often assume the setting where an oracle localizer is available. The proposed architecture provides the generative model with strong geometric priors, which enable it to perform localization without needing an oracle and accurate view generation. A toy image generation task and vizdoom navigation experiments demonstrate the improved performance of the proposed model over a previous (non-spatially-structured) generative baseline method in terms of view synthesis quality and localization accuracy. The paper's main novelty can be summarized as augmenting MapNet's localization and registration modules with an integrated generative process. This seems like an important contribution: generating a whole (or parts of a) scene from a prior distribution could have important downstream effects in exploration, agent navigation and memory systems, where an agent can explicitly reason about where objects or rewards likely are in the environment based on past experience. This is in stark contrast to the current standard for unexplored areas in spatial memories, which are either zeroed out or initialized to a single learnable embedding. Additionally, the generative process can be a useful window into the agent's mapping procedure, enabling visualization and rendering of the agent's semantic representation of the scene, which is for the most part opaque to the agent designer for current spatial memories (except in the case where the memory is forced to represent a provided semantics, like in CogMap). Despite the positive and interesting contribution of generative modeling to spatial memories, there are several concerns with the current paper: (1) Perhaps the most obvious is that the quantized orientation limits the current applicability of the method and I don't think that this point is adequately addressed. Doesn't quantized orientation make localization trivial? A local odometry method can probably learn to predict the motion of very long sequences accurately without drift, especially at the quantization levels tested in this paper. (2) With respect to the first point, I think that odometry is not an interesting evaluation setting for what this model can do. It would be far more informative, for example, if you could train the model in a set of mazes with structured goal spaces, and afterwards test how accurately it can imagine reasonable goal locations. You can then also measure how accurately it is predicting the correct goal location as a function of environment explored, etc. This would directly demonstrate the model's capabilities to be used for downstream exploration and navigation tasks, which is where I believe it would be most impactful. (3) More pressing is that the technical writing in the main text is very poor, with critical architectural details omitted and left unexplained, and figures and tables having minimal captions. For example, despite being a central contribution, the generative process is mostly left unexplained. Is it a deterministic autoencoder or some stochastic variant? The only reference I can find to the generative process is Figure 3, which makes a reference to DAE (I assume it is referring to denoising autoencoders?). (4) There is some description of the generative model in the appendix, but even then it is not completely clear. It seems there is no noise in the generative process? Does that mean that there is only a single possible decoded view given an encoded map feature? If that is the case, this is a serious limitation and seems to be in contention with point (d) in the abstract. In conclusion, the method seems like an otherwise important contribution to spatial memories, but currently poor technical writing, a relatively uninformative evaluation setting and confusing architectural description make me somewhat cautious in recommending this paper for acceptance.

Reviewer 2



Response to author feedback: Thank you for your answers and additional experiments. As a result of them, I have increased my score to 7. ------------------------------------ This paper introduces a novel model for incremental scene synthesis, that uses a location network (Mapnet) and a memory architecture that can be used to reconstruct images at arbitrary viewpoints. Incremental scene synthesis is an interesting and relevant application, that can benefit downstream tasks in many subfields. The proposed model is a complex architecture, but the authors are good at providing a quite clear step-by-step explanation (part of it is in the appendix due to space constraints in the NeurIPS format). It is however less clear to me for which applications there is a strong need for being able to produce hallucinations. Hallucinations are in fact meaningful only in less interesting cases for which it is possible to learn a very strong prior for the images (e.g. experiment with faces in CelebA). However, for many environments (e.g. experiments with the floor plans in the HoME dataset) observing part of the image does not tell you too much about other parts of it, and hallucinations are far from accurate. To my knowledge this is among the first model that is able to perform coherent localization and incremental scene synthesis in an end-to-end fashion, and the experimental results look convincing. They could however be further improved to provide a deeper understanding of the real competitiveness of the model. The GTM-SM model that the authors use in the experiments is a relevant competing method, that focuses however on a fairly different task, and is therefore hard to provide meaningful comparison keeping it in its original form. The GTM-SM is built to be able to get long-term action-conditioned predictions for planning application in RL, not scene registration/synthesis (although it can do it to a certain extent). This model in fact is trained in an unsupervised way, feeding action and image information during training, and not location information as in the proposed model. However, the learned state-space model in the GTM-SM could also greatly benefit from the location information (at least during training) that is assumed to be available in the experiments in this submission. For this I believe ground-truth locations should be also used in the GTM-SM experiments to provide a more fair comparison (it is true that the GTM-SM has also access to additional action information in the experiments, but this information is far weaker than the location information considering that the GTM-SM needs to learn a transition model). You may also consider passing the viewpoints to the GTM-SM as well when generating images (the model basically becomes a GQN), which would certainly give an unfair advantage to the GTM-SM, but its results would be very helpful to assess the potential of the proposed architecture. I am quite concerned about the correctness of your GTM-SM implementation. Considering the results in the original paper [8] it seems to me that a well specified and trained GTM-SM should perform much better, at least in the anamnesis metric in the A_cel^s experiment, but likely also in other experiments. Could you provide more details on GTM-SM implementation and training procedure? For the proposed model to work there has to be lots of correlation and overlapping features among the initial observation to be able to learn to correctly localize and memorize. How can this model be used for efficient exploration in larger environments?

Reviewer 3



Overall, the introduced method is novel and the results it allows generating are interesting. A number of aspects of the work remain unclear in its current state. However, I think these can be addressed as part of the final version of the paper. Therefore, I am leaning towards accepting this work. - L101: What is meant by 'frustum culling' in the context of extracting features from spatial memory? The notion of 'culling' is introduced in Section 3.2 in that it refers to culling features. But as this is part of the main contribution it should be made more clear what is meant by this. Furthermore, in Section 3.2 it is not clear what the culling operation is really doing. Instead, the description of Equation (3) is stating that the requested neighborhood is filled. So apart from Equation (3) there is no description of what is actually meant by the term 'culling'. This should be clarified. - The description of 'Encoding Memories' is not clear. In lines 112-122 the text states that a sequence of observed images as either RGB or RGB-D is used. Later an example is provided for constructing the agent's spatial memory for RGB-D observations by converting the depth maps into point clouds and existing solutions are presented how to approximate ground planes. - L128: What is a 'patch' in this context? Later the text also refers to 'feature patch'. It should be more formally defined. - Figure 4 should have a more meaningful caption. As is it is not comprehensible what is illustrated this figure. - L159: 'field of view' is the more common term. - When I am not mistaken, 's' and 't' (in Equations 3, 4) are not defined (also not in the supplementary material). - L208: What is the unit of 83x83? - L216: What is a patch here?

[Author Response · NeurIPS 2019]

We are grateful to all the reviewers for the insightful comments and suggestions. We are delighted that the **novelty** and
**efficacy** of our method, as well as its potential to stimulate further research, have been acknowledged by all.

**[R1] Technical writing.** We are very sorry for any confusion our writing may have caused and are grateful to R1 for
the suggestions. We read through the paper in its entirety and have identified areas where we will improve our writing
(*e.g.*, deep auto-encoders (DAEs) with noise and the culling step in Sec. 3.2-3.3). We will clarify in the final version.

**[R1] Quantization.** We note quantization in our method is an application-specific design choice rather than a limitation.
When compute power and memory allow, finer quantization can be used to obtain better localization accuracy (see
comparison to ORB-SLAM and others in [11]). In our case, relatively coarse quantization is sufficient for scene
synthesis, where the global scene representation is more crucial. We will conduct more experiments with different
quantization levels (experiments running as we write this) and include results in the final version.

**[R1] Evaluation setting & downstream benefits.** We evaluated on navigation and exploration tasks using standard
metrics (*e.g.*, trajectory and position errors as in MapNet). We thank R2 for the suggested maze experiment which we
are currently conducting. In the meantime, results for a similar experiment, measuring the predicted global scene quality
(SSIM) w.r.t. agent steps & fraction of scene observed, are in Figure I (b). Please note the improved SSIM over time.

**[R1] Stochastic Hallucinations.** Our method can be used with or without noise input depending on the application
(*e.g.*, no noise for deterministic navigation planning or heavy noise for image dataset augmentation). Figure I (a) shows
how our proposed model predicts different global properties for identical trajectories with different hallucinatory noise
(with convergence as observations accumulate). We apologize for the confusion and will clarify the DAE architecture.

**[R2] GTM-SM Comparison.** We used a GTM-SM implementation available on GitHub and were able to reproduce
results reported in the paper. We thank R2 for suggesting comparison with custom versions of GTM-SM with pose
information. We show results in Table I. While the performance of these variants is much improved (when compared to
those in our submission), we still observe the superiority of our method w.r.t. prediction of unseen regions.

**[R2] Hallucination Benefits.** R2 is right that hallucinations are more reliable when target scenes have learnable priors
(*e.g.*, structure of faces). Hallucination of uncertain content can be of lower quality due to the trade-off between
representing uncertainties w.r.t. missing content and unsure localization (giving blurred results), and synthesizing
detailed (but likely incorrect) images. Soft registration and hallucinations' statistical nature can add "uncertainty",
whereas our generative components partially compensate for them (*e.g.*, our choice of GAN to improve sampling). For
data generation use-cases, relaxing hallucination constraints and scaling up $\mathcal{L}_{hallu}$ & $\mathcal{L}_{anam}$ can improve image detail,
at the price of possible memory corruption (we focused on consistency rather than high-resolution hallucinations).

**[R2] Larger Environments.** One interesting direction for future work could be the use of pyramidal/multi-scale
memory maps for refined registration/synthesis or for capturing larger scenes.

**[R3] Feature Culling.** We are very sorry for unclear exposition (some explanation is in supp. material). Inspired by
*culling* in computer graphics, our process extracts features from memory to sample the view from a requested viewpoint,
ignoring features (-1 in Eq. 3) outside the agent's field of view at that position. We will clarify in the final version.

**[R3] Unclear Terms.** We thank R3 for the suggestions and will include them in the final version (Fig. 4 caption,
replacing "patch" & "view field", "$83 \times 83$px", *etc.*). Please note $t$ is defined L61 and L111 and $s$ in L114 and L159.

Table I: Comparison w.r.t. $A_{cel}^s$, editing GTM-SM to leverage ground-truth locations $l_t$.

| Methods | Average Position Error | | | Absolute Trajectory Error | | | Anam. Metr. | | Hall. Metr. | |
|---|---|---|---|---|---|---|---|---|---|---|
| | Med.↘ | Mean↘ | Std.↘ | Med.↘ | Mean↘ | Std.↘ | L1↘ | SSIM↗ | L1↘ | SSIM↗ |
| GTM-SM trained with L1 loss between $s_t$ and $l_t$ | **1.0px** | 1.03px | 1.23px | 0.79px | 0.87px | 0.86px | 0.13 | 0.64 | 0.15 | 0.40 |
| GTM-SM fed with $l_t$ as $s_t$ (no localization) | 0px (NA – poses passed as inputs) | | | 0px (NA – poses passed as inputs) | | | 0.08 | 0.76 | 0.13 | 0.43 |
| Ours | **1.0px** | **0.68px** | **1.02px** | **0.49px** | **0.60px** | **0.64px** | **0.06** | **0.80** | **0.09** | **0.72** |

Figure I: (a) Global sampling for the same trajectories but different noise passed to the hallucinatory DAE at each step $t$; (b) SSIM of the global scene representation w.r.t. agent steps and scene observed for $A_{cel}^s$.

[Meta-Review · NeurIPS 2019]

This submission initially received positive but cautious reviews, with two reviewers recommending a borderline accept. After reading the author response, both of these reviewers were satisfied that their concerns were addressed (regarding correctness of GM-SM implementation + the technical writing), so they both increased their score to "accept." While more experiments would have made the paper stronger, all reviewers ultimately agreed that this is an important problem and the paper makes a good contribution that can spur future work in the area.